# Parkinson’s Disease Symptoms Associated with Developing On-State Axial Symptoms Early after Subthalamic Deep Brain Stimulation

**DOI:** 10.3390/diagnostics12041001

**Published:** 2022-04-15

**Authors:** Gustavo Fernández-Pajarín, Ángel Sesar, José Luis Relova, Begoña Ares, Isabel Jiménez, Miguel Gelabert-González, Eduardo Arán, Alfonso Castro

**Affiliations:** 1Department of Neurology, Hospital Clínico Universitario de Santiago, 15706 La Coruña, Spain; angel.sesar.ignacio@sergas.es (Á.S.); begona.ares.pensado@sergas.es (B.A.); isabel.jimenez.martin@sergas.es (I.J.); alfonso.castro@usc.es (A.C.); 2Instituto de Investigación Sanitaria de Santiago de Compostela (IDIS), 15706 La Coruña, Spain; eduardo.aran.echabe@sergas.es; 3Department of Clinical Neurophysiology, Hospital Clínico Universitario de Santiago, 15706 La Coruña, Spain; joseluis.relova@usc.es; 4Department of Neurosurgery, Hospital Clínico Universitario de Santiago, 15706 La Coruña, Spain; miguel.gelabert@usc.es

**Keywords:** Parkinson’s disease, advanced state, deep brain stimulation, freezing of gait, axial symptoms, early axial symptoms, low-frequency stimulation, high-frequency stimulation

## Abstract

Background: The relationship between axial symptoms in Parkinson’s disease (PD) and subthalamic deep brain stimulation (STN-DBS) is still unclear. Purpose: We searched for particular clinical characteristics before STN-DBS linked to on-state axial problems after surgery. Methods: We retrospectively analyzed baseline motor, emotional and cognitive features from PD patients with early axial symptoms (within 4 years after STN-DBS) and late axial symptoms (after 4 years). We also considered a group of PD patients without axial symptoms for at least 4 years after surgery. Results: At baseline, early-axial PD patients (*n* = 28) had a higher on-state Unified Parkinson’s Disease Rating Scale III (15.0 ± 5.6 to 11.6 ± 6.2, *p* = 0.020), higher axial score (2.4 ± 1.8 to 0.7 ± 1.0, *p* < 0.001) and worse dopaminergic response (0.62 ± 0.12 to 0.70 ± 0.11, *p* = 0.005), than non-axial PD patients (*n* = 51). Early-axial PD patients had short-term recall impairment, not seen in non-axial PD (36.3 ± 7.6 to 40.3 ± 9.3, *p* = 0.041). These variables were similar between late-axial PD (*n* = 18) and non-axial PD, but late-axial PD showed worse frontal dysfunction. Conclusions: PD patients with early axial symptoms after DBS may have a significantly worse presurgical motor phenotype, poorer dopaminergic response and memory impairment. This may correspond to a more severe form of PD.

## 1. Introduction

Parkinson’s disease (PD) is the second neurodegenerative disease in frequency after Alzheimer’s disease. On-axial symptoms in Parkinson’s disease (PD) include gait, posture and balance problems, dysarthria and dysphagia. They are frequent in the advanced stage of the disease [1]. Given its non-dopaminergic nature, there is currently no effective therapy. Therefore, its presence represents a significant loss of independence for the patient [2]. Subthalamic deep brain stimulation (STN-DBS) consists of implanting an electrode in each subthalamic nucleus, hyperactive in PD. Electric stimulation of these electrodes reduces the STN activity, giving rise to a symptomatic improvement. Subjects with on-axial symptoms are excluded from this technique [3]. Only in some cases may they be treated with infusion therapies [4].

Freezing of gait has been correlated to the dysfunction of the supraspinal locomotor network [5]. This network involves mainly the primary motor cortex, the supplementary motor area, the parietal cortex, the basal ganglia, the STN, the mesencephalic locomotor region and the cerebellum.

PD shows remarkable clinical heterogeneity. Combining motor and non-motor symptoms, different PD phenotypes have been established [6]. Mainly motor, with a younger age at onset, mild motor symptoms and good dopaminergic response. Diffuse malignant, with variable age of onset but generally older, early dopaminergic-resistant motor symptoms and a significant burden of non-motor symptoms at onset. A third group was established, called Intermediate, for those not classifiable individuals.

These PD clinical subtypes, already identifiable in early phases of the disease, are to true subgroups within PD rather than to different phases of the same entity. Recently, it has been suggested that different PD clinical subtypes may have different speeds in the pathological process until they reach a common endpoint [7]. Previous clinicopathological studies supported a staging system based on the rostral extent and severity of Lewy body pathology according to age [8] or levodopa response [9]. In addition, it has been demonstrated that non-tremor patients had more severe neocortical pathology [10], accompanied by Alzheimer-like pathology and vascular changes [11].

Gait is not affected at initial stages of PD. Later, its deterioration implicates a degeneration of non-dopaminergic pathways, maybe cholinergic [12]. Cognitive dysfunction in PD was linked to slow gait speed [13,14]. This clinical heterogeneity of PD reflects the involvement of different brain systems, not only a diminished dopaminergic state. In some instances, axial symptoms occur early after STN-DBS surgery. The cause of this is still a matter of debate. Recent studies seem to support that STN-DBS has no negative long-term effect on axial symptoms [15,16,17].

Nevertheless, these studies curiously compare axial symptoms in *off meds* before STN-DBS with *on stim*. This only shows the STN-DBS beneficial effect on the *off*-state axial symptoms. Off-state axial symptoms are mainly due to bradykinesia and rigidity and respond to dopaminergic treatments. 

On the other hand, some groups have described improvement in axial symptoms by switching to low-frequency stimulation [18,19]. This has been controversial. Indeed, axial symptoms have encouraged to search for other surgical targets for DBS, such as the pedunculopontine nuclei (PPN) [20,21], or simultaneous low-frequency stimulation of substantia nigra *pars reticulata* and high-frequency stimulation of subthalamic nuclei (STN) [22,23].

*On*-state axial symptoms are currently poorly understood and many times, they have been approached in somewhat controversial manners. This study aims to search for a possible presurgical parkinsonian profile associated with the development of *on*-state axial symptoms after surgery.

## 2. Patients and Methods

This retrospective and observational study conducted on a registry of PD patients with STN-DBS included consecutively to our maintained databank from the Movement Disorder Unit of the Hospital Clínico Universitario de Santiago de Compostela, Spain. For determining the criteria for STN-DBS, we have based on the Core Assessment Program for Surgical Interventional Therapies in Parkinson’s Disease (CAPSIT-PD) [24] with some modifications. These were the criteria: (1) advanced-stage PD, (2) disease duration over five years, (3) reduction in Unified Parkinson’s Disease Rating Scale III (UPDRS III) over 50% after levodopa (LD) trial or apomorphine challenge test, (4) age under 71, (5) MRI with no significant vascular damage or structural abnormalities, (6) absence of significant cognitive decline according to selected neuropsychological scales, (7) lack of severe psychiatric conditions, except drug-induced psychosis, (8) absence of on-time major gait problems, (9) good general health, and (10) realistic expectations.

We recorded demographical (age at PD diagnosis, age at STN-DBS, sex, medication and Levodopa Equivalent Daily Dose -LEDD- and medical history for hypertension or diabetes) and clinical data. We assessed daily patient activities and motor state with the UPDRS part II and part III, resulting from a preoperative evaluation around six months before the surgery. After the overnight withdrawal of dopaminergic medications and after a levodopa challenge with 1.5 times the morning dose, we evaluated the patients. Apart from the total motor score of UPDRS part III, we individually analyzed an axial score (speech, rising from a chair, posture, gait and stability) and a tremor score (rest tremor in upper limbs, tremor in lower limbs and postural or kinetic tremor in upper limbs) in *on*-state, as well as UPDRS part II items for falls, freezing and walking. We calculated UPDRS improvement (i) as follows: UPDRS III *(i)* = UPDRS III *off*-UPDRS III *on*/UPDRS III *off*.

When performing an STN-DBS surgery, all patients had a neuropsychological evaluation. We evaluated the global cognitive status with the Mattis Dementia Rating Scale (MDRS). Memory was assessed using the Rey Auditory-Verbal Learning Test (RAVLT). Executive functions were evaluated by 3-piece Tower of Hanoi. Working memory was measured the Wechsler Adult Intelligence Scale (WAIS-digits) digits test digits). Visual memory and visuospatial skills were evaluated using the Benton Visual Retention Test (BVRT) and the Benton Judgement of Line Orientation (BJLO). Verbal phonetic “*p*” and category “animal” fluency were measured. The Yesavage Depression Scale (YDS) was performed to quantify depression (>14) or subdepressive mood (>11).

### Statistical Analysis

Data are expressed as percentages for qualitative variables and mean and standard deviation for quantitative variables (we have included UPDRS and items or derived scores in this group). Data normality and equality of variances were assessed using the Shapiro–Wilk and Levene tests. ANOVA for repeated measures was completed for the quantitative variables to determine statistical differences across three situations (non-axial, early-axial and late axial-PD) if the normal distribution assumptions were met. If not, the Friedmann test was used. Post hoc analyses were completed using *t*-tests with Bonferroni correction. The Cochran test was chosen to compare more than two groups for categorical variables.

## 3. Results

Between January 2010 and December 2020, a total number of 170 PD patients underwent STN-DBS surgery. For the follow-up analysis, sixteen patients were excluded due to the following reasons: nine had their follow-up elsewhere, four had intolerability to levodopa, so we lacked an *on meds* evaluation, and one had an acute complication with permanent neurological sequelae (deep intracranial hemorrhage). We did not consider either two patients who developed unmistakable symptoms of atypical parkinsonism and one patient who suffered from cervical myelopathy, resulting in a permanent tetraparesis. 

For the remaining patients (*n* = 153), we selected those who experienced a significant gait, balance and speech impairment, defined by 8 or more points in the axial score [25,26,27], in at least two consecutive clinical evaluations. We divided the axial complication into occurring within four years after surgery (early-axial PD) and beyond four years (late-axial PD). We established the third group with patients with no axial symptoms after four years (non-axial PD). All patients selected as *axial-PD* had a negative levodopa challenge with stimulation *off* (see below in discussion, limitations paragraph).

We have identified 46 PD patients who experienced relevant and limiting axial symptoms in their evolution since surgery. We allocated 28 patients in early-axial PD (mean time 28.1 ± 12.5 months) and 18 in late-axial PD (75.3 ± 19.2 months). Fifty-one patients were included in non-axial PD (Figure 1).

At baseline presurgical evaluation, early-axial PD was significantly older than non-axial PD (62.5 ± 5.6 to 58.3 ± 8.7, *p* = 0.010), but not older than late-axial PD. All three groups did not differ in years with PD before surgery or in LEDD. Early-axial PD had significantly higher scores for *on*-state UPDRS II (8.1 ± 3.9 to 5.5 ± 4.1, *p* = 0.008), including gait (1.0 ± 0.5 to 0.4 ± 0.6, *p* < 0.001), in comparison to non-axial PD. Early-axial PD also had significantly higher scores in *on*-state for UPDRS III (15.0 ± 5.6 to 11.6 ± 6.2, *p* = 0.020), axial score (2.4 ± 1.8 to 0.7 ± 1.0, *p* < 0.001) and lower UPDRS improvement (0.62 ± 0.12 to 0.70 ± 0.11, *p* = 0.005) in comparison to non-axial PD. Concerning late-axial PD, early-axial PD only reached statistical differences for freezing and falls. Late-axial PD had motor scores similar to non-axial PD, except for a lower tremor score (0.3 ± 0.6 to 0.8 ± 1.3, *p* = 0.022). Clinical characteristics before STN-DBS of each group are detailed in Table 1.

At baseline, early-axial PD had significantly lower scores for RAVLT short-term recall (36.3 ± 7.6 to 40.3 ± 9.3, *p* = 0.041) concerning non-axial PD. Besides, early-axial PD had the lowest scores of all three groups in RAVLT long-term recall and BJLO, although there were no statistical differences between them. Late-axial PD had significantly lower scores at baseline for MDRS (130.3 ± 7.5 to 134.4 ± 6.2, *p* = 0.025), BVRT (10.1 ± 2.6 to 11.7 ± 2.1, *p* = 0.013) and 3-piece Tower of Hanoi (72.2 to 96.1%, *p* = 0.010) in comparison to non-axial PD. Early-axial and non-axial PD had similar scores for global cognition (MDRS) and frontal tasks (3-piece Tower of Hanoi, verbal fluency and WAIS-digits), and late-axial PD non-axial PD did not differ so much in memory. Early-axial PD had higher scores for YDS obtaining results for established depression. Neuropsychological evaluation before DBS of each group is shown in Table 2.

## 4. Discussion

In this study, early-axial PD showed some motor and cognitive characteristics. Regarding motor profile, we found higher UPDRS III and axial scores in on-state and lower UPDRS improvement as the main predictors for developing prominent early-axial symptoms in the STN-DBS PD patients (Figure 2). This previous worse situation in gait and posture may represent a more severe course of the disease [28,29], with a considerable non-dopaminergic component. A weaker dopaminergic response also supports this. In contrast, non-axial and late-axial PD show better scores and are very similar.

The profile of cognitive impairment in PD is usually frontal or frontosubcortical, with a predominant executive and visuospatial dysfunction and relatively unaffected memory [30]. A common claim is that the memory deficiency in PD is of retrieval rather than encoding and storage. Nevertheless, there is evidence for verbal and non-verbal memory deficiency in dementia associated with Parkinson’s Disease (PDD) [31]. In our study, early-axial PD has significantly lower scores for verbal memory, while late-axial PD did worse in frontal tasks and visual memory (Figure 3). At the same time, early-axial PD has similar ratings in frontal tasks and late-axial PD in verbal memory compared to non-axial PD. This could represent two different cognitive profiles. Early-axial PD would have features for temporoparietal involvement and late-axial PD would display elements for a more “classical” frontal and posterior cortical dysfunction.

It was identified that gait is prone to decline when certain specific cognitive domains are impaired. Gait is controlled by both cortical (conscious and slow) and subcortical (automatic and fast) networks. PD patients are compelled to rely on cortical control due to basal ganglia dysfunction [32]. A decline in attention, mostly dopaminergic and cholinergic, was related to a slower pace, higher gait variability, and unstable postural control. The decline in visual memory, cholinergic, was also associated with a slower pace [7]. In our series, late-axial PD complies with this observation. With a selective decline in memory, early-axial PD may represent a more severe and distinctive cholinergic loss or even amyloid co-pathology [33]. Cortical cholinergic denervation was found as a marker for gate slowing in PD, and interestingly, the effect was driven by basal forebrain but not by PPN denervation [7].

We may not exclude that non-axial and late-axial PD probably represent a continuum in the disease. Some non-axial PD patients will eventually become late-axial. Nevertheless, our data suggest that late-axial PD represents the classic rigid-akinetic phenotype of the disease. This does not mean a limitation for the analysis of patients who develop early prominent axial symptoms since we have not considered non-axial PD patients under four years after surgery.

*On-axial* symptoms involve a more severe stage of PD, as they do not respond to dopaminergic medication. A meta-analysis [10] and two recent studies [12,13] show a presumed positive effect of STN-DBS on axial symptoms by analyzing the axial items of UPDRS. The methodology, nevertheless, seems controversial. At follow-up, the *off meds/on stim* situation significantly improves axial symptoms compared to the baseline *off meds*. Therefore, the effect of STN-DBS lies in a better stimulus for previously dopaminergic, responsive axial symptoms. Therefore, what happens when the axial symptoms do not respond to the medication remains to be evaluated. Actually, “on-freezing” patients are not candidates for STN-DBS.

When the axial symptoms happen in operated patients, especially if they do it early, we should assess whether they occur in an *off* (worst) or in an *on* (best) situation. A levodopa challenge with DBS switched off is required [34]. An improvement in axial symptoms indicates a dopaminergic response and suggests an ineffective STN-DBS, probably due to misplaced electrodes or an inadequate contact selection. Axial symptoms remaining unchanged means disease progression, affecting non-striatonigral and non-dopaminergic pathways. 

Therefore, we find the results of the publications that support the effectiveness of lowering the frequency of stimulation (between 60 and 100 Hz) in axial symptoms controversial. About half of the subjects in two of these studies have questionable effectiveness of the STN-DBS as the *off meds/on stim* situation does not substantially improve the off *meds/off stim*, in contrast to the *on meds/on stim* [35,36]. It is hard to think about how lowering frequency may improve axial symptoms when the patient gets no benefit from the STN-DBS. Moreover, the clinical evaluation is carried out one hour after modifying the stimulation frequency [36,37]. Changes in stimulation parameters during minutes or hours are not enough to evaluate their effectiveness.

Many factors may be influencing the DBS adjustments. First, to assess the early response to the stimulation, we should be guided by fast responding signs, such as rigidity, the most reliable sign, or tremor. On the contrary, bradykinesia has a latency of hours to days, so it is less useful [38]. Second, the DBS response may be modulated by external factors. Tremor may be modified by anxiety or verbal suggestion [39]. Bradykinesia is influenced by fatigue, patient expectations and may respond to a placebo effect [40,41,42].

Our clinical experience after over 300 patients operated on in 20 years is in line with observations by Ricchi et al. [25] and Sidiropoulos et al. [26]. The improvement in axial symptoms after lowering the stimulation frequency is transient and very mild concerning a notably worsening of other symptoms of the disease, especially in tremoric patients. We switched back to the previous stimulation parameters in most of our patients after a few days.

This study suggests that different PD phenotypes may have different responses to STN-DBS. To identify these particular phenotypes may be important for predicting the result after surgery. It is remarkable that poor response to stimulation probably has more to do with non-dopaminergic pathways. Current tests for presurgical examination lack sensibility to identify this phenotype with worse result. 

Finally, PD patients who develop earlier prominent axial symptoms show particular motor and cognitive characteristics. On the one hand, they seem to have a previous worse motor phenotype and a poorer response to dopaminergic medication. On the other hand, their cognitive profile is not typical, with selective memory impairment, probably due to a cholinergic dysfunction. There is a proven relationship between cholinergic dysfunction and axial symptoms. PD patients who develop later axial symptoms have similar features to non-axial patients, except for a more pronounced frontal dysfunction. This suggests that longer evolution for PD is critical for developing axial symptoms in late axial PD. We believe that these factors are the main determinants for developing axial symptoms, not the STN-DBS itself, but many aspects of the functionally of cerebral oscillatory networks remain to be elucidated.

### Limitations

Our study has some limitations, first of all, being retrospective. It was chosen to view the mean axial score obtained in the studies cited, in which axial symptoms after STN-DBS were evaluated. To consider the axial symptoms only as a cut-off value of the UPDRS scale may be too simple. In the same way, we selected 4 years as a reasonable lapse of time in the advanced PD. An ongoing study with survival analysis using the time to axial features may provide better data. A similar study in a different site or a single study with data from two sites would be more explanatory. It would be interesting to include the electrodes locations and stimulation parameters employed or medication adjustments. We have always checked the levodopa response when axial symptoms occur in an operated patient. This has ensured its non-dopaminergic nature or makes us suspect of inefficient stimulation.

It is likely that a particular clinical PD phenotype and, therefore, an underlying pathology, are more prone to develop axial symptoms after STN-DBS. However, there may be other factors such as age, vascular comorbidity or changes in medication. There remain many questions to be solved regarding high-frequency stimulation and axial motor symptoms. Two meta-analysis concluded that the benefits of STN-DBS for axial symptoms showed a decline in the long term [43,44]. This has been called “long-term DBS syndrome”, a phenotype comprising relatively well-controlled bradykinesia, rigidity and tremor, but increasing axial motor problems [45,46]. It is also unknown whether high-frequency stimulation conditions an ulterior response when readjusting to low-frequency stimulation or, even, low-frequency stimulation could require a different location of the electrodes to be more effective [27,37]. High-frequency stimulation with high voltages, as frequently happens with patients years after being operated, would be deleterious by spreading the current to nearby structures. Although STN and PPN are not close anatomically, they are functionally interconnected. Since PPN is crucial for gait, low-frequency stimulation of STN has been proposed to alleviate axial symptoms, improving the PPN output by reducing the inhibitory influence of GPi. Unfortunately, this has not shown precise results [21].

## Figures and Tables

**Figure 1 diagnostics-12-01001-f001:**
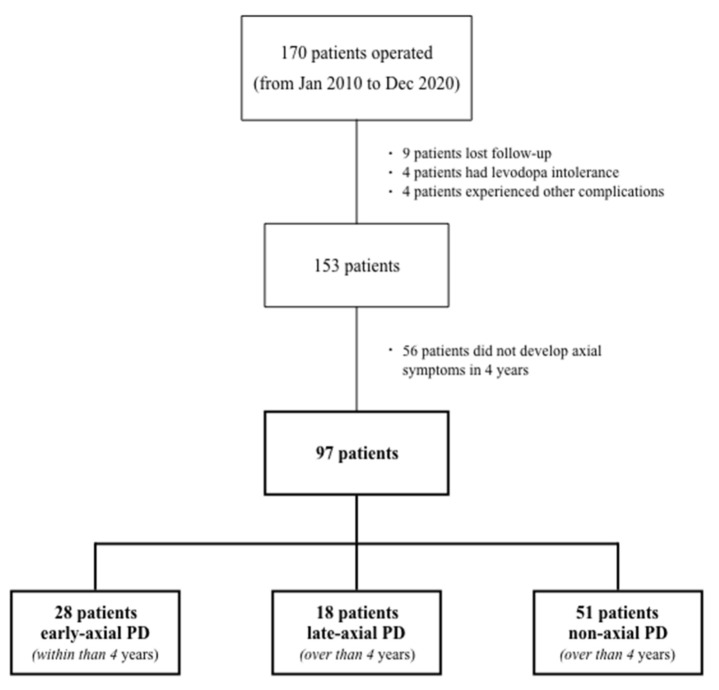
Patients selection criteria. Out of the 153 operated patients followed up for over four years, we discarded those not developing axial symptoms within the first four years. We divided the rest into early-axial, late-axial and non-axial (see text).

**Figure 2 diagnostics-12-01001-f002:**
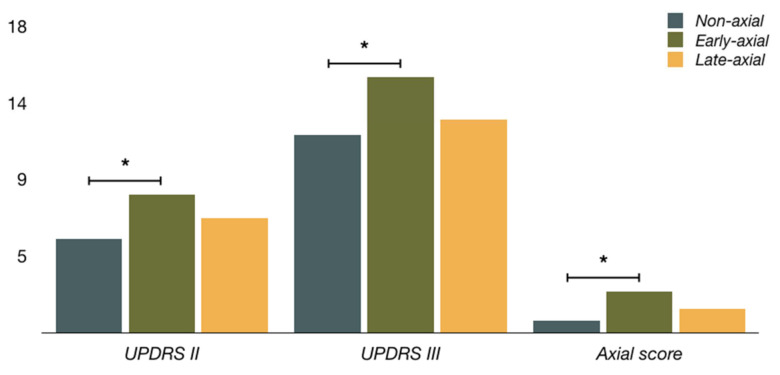
Motor evaluation for every three groups. (UPDRS II—Unified Parkinson’s Disease Rating scale II; UPDRS III—Unified Parkinson’s Disease Rating scale III). * Statistical significance (*p* < 0.05).

**Figure 3 diagnostics-12-01001-f003:**
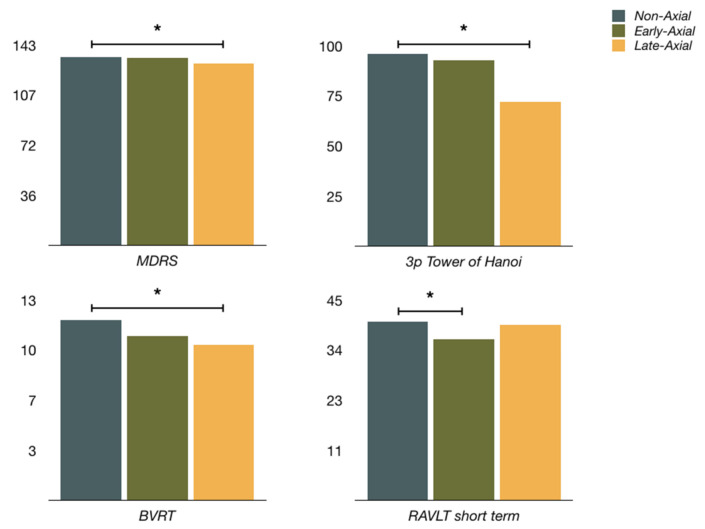
Neuropsychological evaluation for every three groups. (MDRS—Mattis Dementia Rating Scale; RAVLT—Rey Auditory-Verbal Learning Test; BVRT—Benton Visual Retention Test). * Statistical significance (*p* < 0.05).

**Table 1 diagnostics-12-01001-t001:** Baseline clinical characteristics of STN-DBS patients. (UPDRS II—Unified Parkinson’s Disease Rating scale II; UPDRS III—Unified Parkinson’s Disease Rating scale III; LEDD—Levodopa Equivalent Daily Dose; m—months; y—years). ^a^ Significant difference between early-axial and non-axial PD; *p* < 0.05. ^b^ Significant difference between early-axial and late-axial PD; *p* < 0.05. ^c^ Significant difference between late-axial and non-axial PD; *p* < 0.05.

	Non-Axial PD (*n* = 51)	Early-Axial PD (*n* = 28)	Late-Axial PD (*n* = 18)
Sex (male)	24 (47.1%)	16 (57.1%)	10 (55.6%)
Hypertension	10 (19.6%)	6 (21.4%)	5 (27.8%)
Age at STN-DBS ^a^	**58.3 ± 8.7**	**62.5 ± 5.6**	61.1 ± 8.7
Time since PD diagnosis (y)	10.2 ± 3.6	9.9 ± 4.7	10.6 ± 3.3
LEDD (mg)	1348 ± 495	1432 ± 337	1485 ± 600
UPDRS II *on meds* ^a^	**5.5 ± 4.1**	**8.1 ± 3.9**	6.7 ± 3.4
Falls	0.1 ± 0.4	0.3 ± 0.6	0.2 ± 0.5
Freezing ^b^	0.2 ± 0.5	**0.5 ± 0.6**	**0.1 ± 0.3**
Gait ^a,b^	**0.4 ± 0.6**	**1.0 ± 0.5**	**0.6 ± 0.5**
UPDRS III *off meds*	37.7 ± 11.3	40.4 ± 11.6	40.4 ± 9.9
UPDRS III *on meds* ^a^	**11.6 ± 6.2**	**15.0 ± 5.6**	12.5 ± 4.9
UPDRS III (i) ^a^	**0.70 ± 0.11**	**0.62 ± 0.12**	0.69 ± 0.12
Tremor score ^c^	**0.8 ± 1.3**	0.8 ± 2.0	**0.3 ± 0.6**
Axial score ^a^	**0.7 ± 1.0**	**2.4 ± 1.8**	1.4 ± 1.9
Time to axial PD (m)	-	28.1 ± 12.5	75.3 ± 19.2
Follow-up (m)	75.0 ± 19.7	72.9 ± 29.9	100.1 ± 17.4

**Table 2 diagnostics-12-01001-t002:** Baseline neuropsychological evaluation of STN-DBS patients. (MDRS—Mattis Dementia Rating Scale; RAVLT—Rey Auditory-Verbal Learning Test; BVRT—Benton Visual Retention Test; BJLO—Benton Judgement of Line Orientation; WAIS—Wechsler Adult Intelligence Scale; YDS—Yesavage Depression Scale). * Number of patients who made the test correctly (<240 s, <40 movements). ** Cases lost to poor understanding (BVRT, BJLO, WAIS). Test added in 2012 (YDS). ^a^ Significant difference between non-axial and early-axial PD; *p* < 0.05; ^c^ Significant difference between non-axial and late-axial PD; *p* < 0.05.

	Non-Axial PD (*n* = 51)	Early-Axial PD (*n* = 28)	Late-Axial PD (*n* = 18)
MDRS **^c^**	**134.4 ± 6.2**	134.1 ± 5.5	**130.3 ± 7.5**
RAVLT (short-term recall) ^a^	**40.3 ± 9.3**	**36.3 ± 7.6**	39.5 ± 10.6
RAVLT (long-term recall)	8.3 ± 3.4	7.1 ± 2.9	7.7 ± 3.4
3p Tower of Hanoi *^,c^	**49 (96.1%)**	26 (92.9%)	**13 (72.2%)**
BVRT **^,c^	**11.7 ± 2.1** (*n* = 49)	10.7 ± 2.7 (*n* = 27)	**10.1 ± 2.6** (*n* = 18)
BJLO **	24.0 ± 4.1 (*n* = 46)	22.2 ± 6.1 (*n* = 26)	22.7 ± 5.8 (*n* = 17)
WAIS-digits **	8.8 ± 2.8 (*n* = 42)	9.3 ± 4.3 (*n* = 25)	7.1 ± 2.9 (*n* = 16)
Phonetic Fluency	13.9 ± 6.4	12.6 ± 4.7	11.9 ± 5.2
Semantic Fluency	18.9 ± 6.9	16.8 ± 4.7	16.4 ± 6.0
YDS **	10.9 ± 6.1 (*n* = 33)	14.5 ± 7.4 (*n* = 19)	13.6 ± 8.7 (*n* = 5)

## Data Availability

Not applicable.

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
