# Peer review of "Parkinson’s Disease Symptoms Associated with Developing On-State Axial Symptoms Early after Subthalamic Deep Brain Stimulation"

_diagnostics, 2022, doi:10.3390/diagnostics12041001_

Round 1

Reviewer 1 Report

The authors have improved the paper. I do think H&Y staging is a quite relevant feature that should be mentioned in the text and/or in the Table.

Author Response

We thank the reviewer for his/her comments. Actually, we do not have that value for each patient, but all of them were HY2 or 3. We have sent a new manuscript with all the changes highlighted. 

Reviewer 2 Report

17 March 2022

Review on the manuscript titled “Parkinson's disease symptoms are associated with developing on-state axial symptoms early after subthalamic deep brain stimulation” by Fernandez-Pajarin G et al., submitted to Diagnostics

Manuscript ID: diagnostics-1655975

Dear Authors,

The authors have clarified several of the questions raised in the previous round of review. Indeed, they have made some adjustments in order to support the article’s rationale, adding information about neuroanatomical background of PD symptoms, though without taking into account some of the literature suggestions. Providing more information would allow to enrich and complete the theoretical framework as well as deepen the subject of their manuscript, as the bibliography is still too concise. As I stated in my previous review, the abstract should be reorganized to proportionally present the background, purpose, methods, results, and conclusion; also, I deem it unlikely that the issue regarding a more defined background on aetiology and pathogenesis of Parkinson’s disease (PD) can be solved merely by a few and hasty added references. The authors need to open the introduction with more informatively but briefly presenting aetiology, pathology, diagnosis, current treatment, and its challenge. It also deserves describe the heterogeneity and comorbidity of PD, leading to the main topic of this study (https://doi.org/10.3390/ijms22168726; https://doi.org/10.3390/ijms21072431; https://doi.org/10.3390/jpm12010089). Furthermore, I believe that more information on motor and non-motor symptoms (NMS) in PD (that include impairments in specific components of emotional processes and autonomic dysfunctions), would be necessary to truly provide a more thorough analysis on the negative impact of neuropsychiatric, cognitive, autonomic, and sleep complications in PD: in this regard, I suggest again to add finding from additional evidence that have focused on this topic (https://doi.org/10.3390/biomedicines10030627; https://doi.org/10.3390/biomedicines9050517; https://doi.org/10.1007/s00221-020-05829-4;  https://doi.org/10.3390/ijms21072431; https://doi.org/10.1038/s41598-021-82223-2).

Finally, although the authors have plenty discussed the future theoretical and methodological avenues in need of refinement and have offered keys to advancing research and understanding the possible application of neural reuse, I suggest adding a proper defined ‘Limitations’ subsection at the end of the manuscript as well, in which they can discuss those characteristics of the selection of studies or methodology of studies reported that impacted or influenced the or interpretation of the results of this review. I hope this time authors would carefully consider my suggestions.

The manuscript contains three figures, two table and 40 references. The manuscript carries important value presenting the relationship between clinical features before STN-BDS and axial symptoms after the treatment. Thus, I recommend this manuscript for publication after major revision. I am always available for other reviews of such interesting and important articles.

Best regards,

Reviewer

Author Response

We thank the reviewer for the comments. We enclose a document with the rebuttals. We have sent a new manuscript with the changes highlighted.

The abstract should be reorganized to proportionally present the background, purpose, methods, results, and conclusion.

We have arranged the abstract following the reviewer’s suggestion

Also, I deem it unlikely that the issue regarding a more defined background on aetiology and pathogenesis of Parkinson’s disease (PD) can be solved merely by a few and hastily added references. The authors need to open the introduction with more informatively but briefly present aetiology, pathology, diagnosis, current treatment, and its challenge. It also deserves to describe the heterogeneity and comorbidity of PD, leading to the main topic of this study (https://doi.org/10.3390/ijms22168726; https://doi.org/10.3390/ijms21072431; https://doi.org/10.3390/jpm12010089).

We insist on our different views on this particular point. Nobody without a proper background in PD and in DBS is to be interested in this study, so we do not see a need to explain what PD and DBS are. Moreover, we cannot see how to fit an explanation including all the topics suggested without diverting the goal of this paper. On the other hand, we are afraid that the references proposed are too specific to be cited here. Nevertheless, we have included some information about clinical heterogeneity in PD.

Furthermore, I believe that more information on motor and non-motor symptoms (NMS) in PD (that include impairments in specific components of emotional processes and autonomic dysfunctions), would be necessary to truly provide a more thorough analysis on the negative impact of neuropsychiatric, cognitive, autonomic, and sleep complications in PD: in this regard, I suggest again to add finding from additional evidence that has focused on this topic (https://doi.org/10.3390/biomedicines10030627; https://doi.org/10.3390/biomedicines9050517; https://doi.org/10.1007/s00221-020-05829-4;  https://doi.org/10.3390/ijms21072431; https://doi.org/10.1038/s41598-021-82223-2).

All these proposals are very interesting, but our objective is not to analyse PD complications. We just try to suggest that maybe patients with DBS developing early axial symptoms have a particular phenotype. Again, the references proposed do not seem directly linked with the topic of the study.

Finally, although the authors have plenty discussed the future theoretical and methodological avenues in need of refinement and have offered keys to advancing research and understanding the possible application of neural reuse, I suggest adding a proper defined ‘Limitations’ subsection at the end of the manuscript as well, in which they can discuss those characteristics of the selection of studies or methodology of studies reported that impacted or influenced the or interpretation of the results of this review. I hope this time authors would carefully consider my suggestions.

Following the reviewer’s recommendation, we have added an independent subsection at the end of the study dealing with the limitations, with some references on the topic.

The manuscript contains three figures, two table and 40 references. The manuscript carries important value presenting the relationship between clinical features before STN-BDS and axial symptoms after the treatment. Thus, I recommend this manuscript for publication after major revision. I am always available for other reviews of such interesting and important articles.

Round 2

Reviewer 2 Report

28 March 2022

Review on the manuscript titled “Parkinson's disease symptoms are associated with developing on-state axial symptoms early after subthalamic deep brain stimulation” by Fernandez-Pajarin G et al., submitted to Diagnostics

Manuscript ID: diagnostics-1655975

Dear Authors,

The authors partially revised the manuscript, which is limited to the introduction and the limitation. No more reference is added to support the authors’ view and this study. The reference number of 40 is dramatically low. Thus, I suggest citing at least 60-70 references is optimal for original research paper like this manuscript. I leave the previous review report for the convenience of the authors.

The authors have clarified several of the questions raised in the previous round of review. Indeed, they have made some adjustments in order to support the article’s rationale, adding information about neuroanatomical background of Parkinson’s disease (PD) symptoms, though without taking into account some of the literature suggestions. Providing more information would allow to enrich and complete the theoretical framework as well as deepen the subject of their manuscript, as the bibliography is still too concise. As I stated in my previous review, the abstract should be reorganized to proportionally present the background, purpose, methods, results, and conclusion; also, I deem it unlikely that the issue regarding a more defined background on aetiology and pathogenesis of PD can be solved merely by a few and hasty added references. The authors need to open the introduction with more informatively but briefly presenting aetiology, pathology, diagnosis, current treatment, and its challenge. It also deserves describe the heterogeneity and comorbidity of PD, leading to the main topic of this study (https://doi.org/10.3390/ijms22168726; https://doi.org/10.3390/ijms21072431; https://doi.org/10.3390/jpm12010089). Furthermore, I believe that more information on motor and non-motor symptoms (NMS) in PD (that include impairments in specific components of emotional processes and autonomic dysfunctions), would be necessary to truly provide a more thorough analysis on the negative impact of neuropsychiatric, cognitive, autonomic, and sleep complications in PD: in this regard, I suggest again to add finding from additional evidence that have focused on this topic (https://doi.org/10.3390/biomedicines10030627; https://doi.org/10.3390/biomedicines9050517; https://doi.org/10.1007/s00221-020-05829-4;  https://doi.org/10.3390/ijms21072431; https://doi.org/10.1038/s41598-021-82223-2).

The manuscript contains three figures, two table and 40 references. The manuscript carries important value presenting the relationship between clinical features before STN-BDS and axial symptoms after the treatment.

Best regards,

Reviewer

Author Response

We enclose a Word document

Round 3

Reviewer 2 Report

6 April 2022

Review on the manuscript titled “Parkinson's disease symptoms are associated with developing on-state axial symptoms early after subthalamic deep brain stimulation” by Fernandez-Pajarin G et al., submitted to Diagnostics

Manuscript ID: diagnostics-1655975

Dear Authors,

The authors partially revised the manuscript. The reference number of 47 is dramatically low. Thus, I suggest citing at least 60-70 references is optimal for original research paper like this manuscript. I leave the previous review report for the convenience of the authors.

The authors have clarified several of the questions raised in the previous round of review. Indeed, they have made some adjustments in order to support the article’s rationale, adding information about neuroanatomical background of Parkinson’s disease (PD) symptoms, though without taking into account some of the literature suggestions. Providing more information would allow to enrich and complete the theoretical framework as well as deepen the subject of their manuscript, as the bibliography is still too concise. As I stated in my previous review, the abstract should be reorganized to proportionally present the background, purpose, methods, results, and conclusion; also, I deem it unlikely that the issue regarding a more defined background on aetiology and pathogenesis of PD can be solved merely by a few and hasty added references. The authors need to open the introduction with more informatively but briefly presenting aetiology, pathology, diagnosis, current treatment, and its challenge. It also deserves describe the heterogeneity and comorbidity of PD, leading to the main topic of this study (https://doi.org/10.3390/ijms22168726; https://doi.org/10.3390/ijms21072431; https://doi.org/10.3390/jpm12010089). Furthermore, I believe that more information on motor and non-motor symptoms (NMS) in PD (that include impairments in specific components of emotional processes and autonomic dysfunctions), would be necessary to truly provide a more thorough analysis on the negative impact of neuropsychiatric, cognitive, autonomic, and sleep complications in PD: in this regard, I suggest again to add finding from additional evidence that have focused on this topic (https://doi.org/10.3390/biomedicines10030627; https://doi.org/10.3390/biomedicines9050517; https://doi.org/10.1007/s00221-020-05829-4;  https://doi.org/10.3390/ijms21072431; https://doi.org/10.1038/s41598-021-82223-2).

The manuscript contains three figures, two table and 47 references. The manuscript carries important value presenting the relationship between clinical features before STN-BDS and axial symptoms after the treatment.

Best regards,

Reviewer

Author Response

Reviewer 2 has made some interesting suggestions we have followed, dealing with the heterogeneity of PD and the study limitations. We have also rearranged the abstract at the reviewer's proposal, though we seem not to do what he has asked. But we are afraid we cannot change this article the way the reviewer recommends. We try to convey that there may be a particular PD phenotype more prone to develop axial symptoms after PD. We insist that any reader potentially interested in this paper will have a proper background in PD and DBS. If we describe PD features in detail, we will blur our actual goal when writing this. The same happens with the proposed detailed description of PD symptomatology (motor and non-motor). We do not think these suggestions would enrich the study but probably divert our main objective about a PD profile susceptible to axial symptoms. Moreover, once again, we remark that the references provided by the reviewer have not to do with our article

This manuscript is a resubmission of an earlier submission. The following is a list of the peer review reports and author responses from that submission.

Round 1

Reviewer 1 Report

Not sure authors should state that axial symptoms are dopamine sensitive, in the fact that true axial symptoms of gait and balance typically are not (as stated in the conclusion as well)

Should check math on figure #1 numbers don’t add up, the early axial PD number is incorrect.

Table #1 should be reworked to be more easily understood, as it is difficult to follow, showing statistical signifcance in some realms but with high standard deviation.

Reviewer 2 Report

The manuscript: „ Parkinson's disease symptoms associated with developing on-state axial symptoms early after subthalamic deep brain stimulation" by G Fernández-Pajarín and colleagues analyzed the clinical characteristics before STN-DBS linked to on-state axial problems after surgery and identified that patients with early axial symptoms after DBS have a significantly worse presurgical motor phenotype, poorer dopaminergic response and memory impairment. The manuscript is quite informative and the methodology, results and discussion are nicely composed

After thoroughly going through the manuscript, I have a couple of comments:

  1. The study being retrospective, how accurate was the methodology of extracting clinical characteristics?
  2. Was the sample size statistically sufficient enough to draw the conclusions?
  3. Axial disability has been previously reported to be significantly correlated with individual risk of death after surgery. How was this aspect in present cohort?

Reviewer 3 Report

16 February 2022

Review on the manuscript titled “Parkinson's disease symptoms are associated with developing on-state axial symptoms early after subthalamic deep brain stimulation” by Fernandez-Pajarin G et al., submitted to Diagnostics

Manuscript ID: diagnostics-1595058

Dear Authors,

Little is known about the emergence of axial symptoms after the treatment of subthalamic deep brain stimulation (STN-DBS) in patients with Parkinson's disease (PD). The authors conducted a retrospective study to analyze the link between the clinical features of PD patients before STN-DBS and axial symptoms after STN-BDS treatment. The results showed that early axial PD patients had a higher on-state Unified Parkinson's Disease Rating Scale (UPDRS) III, higher axial score, and worse dopaminergic response than non-axial PD patients, that early axial PD patients had short-term recall impairment, and that late-axial PD patients had worse frontal dysfunction. The authors concluded that presurgical motor dysfunction, memory impairment, and dopaminergic response were significantly worse in PD patients with early axial symptoms after STN-DBS.

Please consider the following:

  1. A graphical abstract summarizing the manuscript is highly recommended.
  2. Page 1, Abstract:
  3. Please present more on background and lead to a rationale of this study.
  4. Please expand the abbreviation “UPDRS III”.
  5. The study assessed emotional state as well. Please mention it.
  6. Page 1, Keywords: Please list up to ten keywords.
  7. Pages 1-2, Introduction:
  8. Please present sufficient background on PD in general shortly including epidemiology, pathogenesis, heterogeneity, current treatment, the rationale indication, and the selection criteria of STN-BDS treatment, success rate, adverse effects, and current knowledge, leading to the rationale of this study.
  9. Also, it deserves to present neuroanatomical background of symptoms assessed in this study. Suggested references: https://doi.org/10.3390/ijms22168726; https://doi.org/10.3390/biomedicines9050517; https://doi.org/10.3390/ECCM-10857; https://doi.org/10.3390/ijms21072431; https://doi.org/10.3390/cells9112476; https://doi.org/10.3390/microorganisms9112281;
  10. Page 3, Figure 1: Please add short description in caption.
  11. Pages 6,7, Figures 2,3: The figures should be presented in the results.
  12. Pages 5-8, Conclusion:
  13. Please discuss the previous studies, weakness or limitation in the present review study in depth, potentials and significance of this review, the ultimate goal, research or knowledge needed to achieve, the future research direction, and the biggest challenge in this goal, among others.
  14. Discussion on cognitive, emotional, and motor symptoms linked to post-surgical axial symptoms based on neuroanatomical views which are to be presented in the introduction surely enriches this manuscript. Suggested references: https://doi.org/10.3390/jpm12010089; doi:10.1038/s41380-021-01326-4; doi:10.1016/j.cortex.2021.01.004; doi:10.1016/j.neubiorev.2021.04.036.
  15. Pages 9,10, References: Please cite more references, preferably more than 50 for original articles.

The manuscript contains three figures, two table and 26 references. The manuscript carries important value presenting the relationship between clinical features before STN-BDS and axial symptoms after the treatment. Thus, I recommend this manuscript for publication after major revision.

Best regards,

Masaru Tanaka, M.D., Ph.D.

Reviewer 4 Report

The authors reported a study about axial symptoms in PD after subthalamic deep brain stimulation. I have some comments to the authors:

1) Please add a proper reference to this sentence “Only in some cases, they may be treated with infusion therapies.”

2) In the first paragraph of the methods, please better define the nature of your study (i.e. delete the “prospectively” part).

3) You should better describe your registry before the analysis that you made, and please do no state that details are explained in other works, it’s important to have all information in this paper. Which are the inclusion criteria for the registry? What are the clinical and demographic features that you collect? Which are the exclusion criteria for your registry? This should part be right after the first paragraph of the methods

4) Please state once the meaning of LEDD (levodopa equivalent daily dose).

5) Axial score and tremor score. Please better define how you have calculated those items and include the references that you have used to define them.

6) At the end of the second paragraph of methods please include one/two sentences describing that you checked all PD patients for functional motor disorders. Recently the topic has gained a lot of attention due to the high frequency of these disorders. I recommend these four references to be added in the text:

Tinazzi M, et al. Functional motor phenotypes: to lump or to split? J Neurol. 2021 Dec;268(12):4737-4743

Tinazzi M, et al. Functional motor disorders associated with other neurological diseases: Beyond the boundaries of "organic" neurology. Eur J Neurol. 2021 May;28(5):1752-1758.

Breen DP, et al. Functional movement disorders arising after successful deep brain stimulation. Neurology. 2018 May 15;90(20):931-932.

Maciel R, et al. Functional Dyskinesias following Subthalamic Nucleus Deep Brain Stimulation in Parkinson's Disease: A Report of Three Cases. Mov Disord Clin Pract. 2020 Dec 2;8(1):114-117.

7) Tables: please add HY stages.

8) You should describe figure 2 and 3 in the results, and then comment the findings in the discussion. The figures have also poor quality, please try to improve it.